# *Nosema locustae* (Protozoa, Microsporidia), a Biological Agent for Locust and Grasshopper Control

**Long Zhang** [1,2,*] **and Michel Lecoq** [3]

1   Shandong Academy of Agricultural Sciences, Jinan 250100, China
2   Department of Grassland Resources and Ecology, College of Grassland Science and Technology, China Agricultural University, Beijing 100193, China
3   CIRAD, UMR CBGP, F-34398 Montpellier, France; mlecoq34@gmail.com
*   Correspondence: locust@cau.edu.cn

**Abstract:** Effective locust and grasshopper control is crucial as locust invasions have seriously threatened crops and food security since ancient times. However, the preponderance of chemical insecticides, effective and widely used today, is increasingly criticized as a result of their adverse effects on human health and the environment. Alternative biological control methods are being actively sought to replace chemical pesticides. *Nosema locustae* (Synonyms: *Paranosema locustae*, *Antonospora locustae*), a protozoan pathogen of locusts and grasshoppers, was developed as a biological control agent as early as the 1980s. Subsequently, numerous studies have focused on its pathogenicity, host spectrum, mass production, epizootiology, applications, genomics, and molecular biology. Aspects of recent advances in *N. locustae* show that this entomopathogen plays a special role in locust and grasshopper management because it is safer, has a broad host spectrum of 144 orthopteran species, vertical transmission to offspring through eggs, long persistence in locust and grasshopper populations for more than 10 years, and is well adapted to various types of ecosystems in tropical and temperate regions. However, some limitations still need to be overcome for more efficient locust and grasshopper management in the future.

**Keywords:** locust; grasshopper; biological control; *Nosema locustae*; application; epizootics



## 1. Introduction

Locust and grasshopper (L&G) outbreaks, often resulting in huge plagues, have been a serious threat to global food security since ancient times [1]. Traditional control of L&G consists mainly of the application of chemical pesticides, which often results in many side effects. These include toxic chemical residues on food, adverse health effects on humans and nontarget animals, and environmental pollution. One of the most promising alternatives to chemical pesticides is biological control. Although there are many natural enemies of L&G [2–4], only a few have been developed as biological control agents or potential agents, including the microsporidian *Nosema locustae* (synonym: *Antonospora locustae*, *Paranosema locustae*) and the fungus *Metarhizium acridum*, both of which have been quite widely used in the control of L&G.

*N. locustae*, a unicellular eukaryote (Figure 1) with an obligate intracellular lifestyle [5–7], was the first to be commercially developed for L&G control since its discovery and characterization of its potential as a biological control agent against these species [8–10]. Due to its slow action and various other constraints, *N. locustae* was considered to be of limited application [11,12]. However, as the only microsporidian agent for L&G control, *N. locustae* has been studied extensively [1,13–17]. In recent years, there has been a renewed interest in *N. locustae*, mainly due to work in China, where it is produced in large quantities and used extensively, and Argentina, where its long-term persistence appears to reduce the frequency and intensity of grasshopper outbreaks [18]. Recent advances in areas such as mass production, formulation, application, and epizootiology have

further promoted the application of this pathogen. Numerous studies on its host spectra, molecular biology, and evolution have provided a solid fundamental basis for the use of *N. locustae* for L&G management.

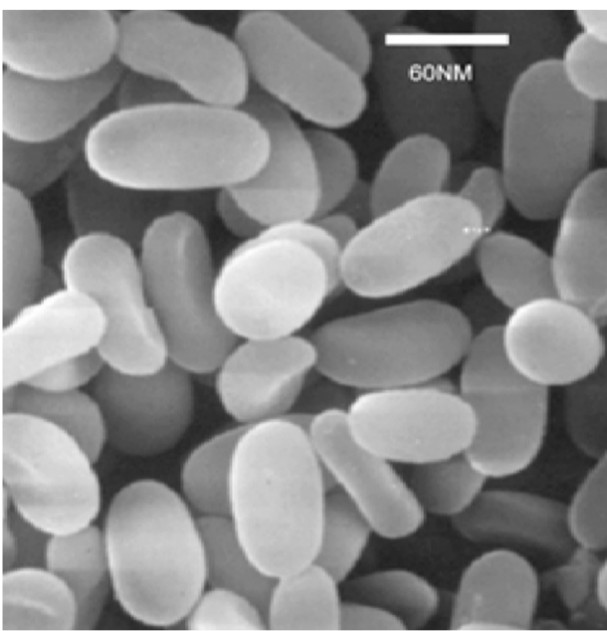

**Figure 1.** Spores of *N. locustae* under scanning electron microscopy (photo by Long Zhang).

## 2. Host Spectrum

*N. locustae* was first identified in a laboratory population from the migratory locust *Locusta migratoria* (Linnaeus, 1758) by Elizabeth Canning in 1953 [19]. This pathogen occurs under natural conditions and has been found in various areas of the United States (Montana, Northern Dakota, Minnesota, Oregon, Wyoming, Colorado, Arizona, and Idaho), Canada (Saskatchewan, Ontario) and South America (western Pampas and northwestern Patagonia, Argentina), as well as Asia (Rajasthan and Vidarbha, India; Inner Mongolia, Hainan and Qinghai, China) and Africa (Karoo, South Africa) [20]. The highest level of infection was reported of about 5.5% in *Melanoplus sanguinipes* (Fabricius, 1798) in Idaho between 1963 and 1967 [21].

*N. locustae* has a wide host range restricted to orthopteran insects. Henry [22] in 1969 provided an initial list of 55 North American species that he knew to be susceptible. Brooks [15] published in 1988 a new list of worldwide susceptible orthopterans, expanding the host range to 95 species. The last review was done in 2005 by Lange [20], who arrived at a total of 121 species. Since then, various authors have been added to this list (Table 1), and we now reach a total of 144 susceptible orthopterans.

**Table 1.** List of orthopteran species infected by *N. locustae* but not included in Lange [20].

| Species | Inoculation Infection (Caging) | Field Trial Infection | References |
|---|:---:|:---:|:---:|
| **Asia** | | | |
| *Bryodemella holdereri* (Krauss, 1901) | X | | [23] |
| *Calliptamus italicus* (Linnaeus, 1758) | X | | [24] |
| *C. abbreviatus* Ikonnikov, 1913 | X | | [23] |
| *Ceracris kiangsu* Tsai, 1929 | X | X | [25] |
| *Chondracris rosea* (De Geer, 1773) | X | X | [26] |
| *Chorthippus dubius* (Zubovski, 1898) | | X | [27] |
| *C. brunneus* (Thunberg, 1815) | X | | |
| *Damalacantha vacca* (Fischer von Waldheim, 1846) | X | | [23] |
| *Deracanthella aranea* (Fischer von Waldheim, 1833) | X | | |
| *Dociostaurus kraussi* (Ingenitskii, 1897) | X | | [24] |
| *Fruhstorferiola tonkinensis* (Willemse, 1921) | | X | [28] |
| *Gampsocleis sedakovii* (Fischer von Waldheim, 1846) | X | | [23] |
| *Haplotropis brunneriana* Saussure, 1888 | X | | |
| *Oedaleus decorus* (Germar, 1825) | X | | [24] |
| *Arcyptera meridionalis* Ikonnikov, 1911 | X | X | |
| *Sphingonotus mongolicus* Saussure, 1888 | X | | [23] |
| **Africa** | | | |
| *Acrotylus blondeli* Saussure, 1884 | | X | |
| *Acrotylus patruelis* (Herrich-Schäffer, 1838) | | X | [29] |
| *Aiolopus thalassinus* (Fabricius, 1781) | | X | |
| *Anacridium melanorhodon* (Walker, 1870) | | X | |
| **America** | | | |
| *Amblytropidia australis* Bruner, 1904 | | X | [30] |
| *Dichroplus vittigerus* (Blanchard, 1851) | | X | |

Although *N. locustae* has a broad host spectrum in Orthoptera, it is incapable of affecting non-orthopteran insects. It has been shown that the honey bee (*Apis mellifera*) and the lepidoterans *Heliothis zea* and *Agrotis ipsilon* were not susceptible [15]. Menapace et al. [31] reported that honey bees were not infected, even when fed high doses of spores. The American cockroach *Periplaneta americana* and the spider *Butalus occidentalis* were also found to be non-susceptible [32]. *N. locustae* has been demonstrated to be safe for vertebrates. Brooks [15] summarized the assessment tests on primary skin irritation, acute dermal toxicity, acute inhalation toxicity and pathogenicity, subacute oral toxicity, acute oral toxicity, acute pathogenicity, and possible hazards of *N. locustae* to vertebrates, including rabbits, guinea pigs, rainbow trout, giant toads, mallards, ring-necked pheasant, mice, and rats. No significant effects were observed. The toxicity of *N. locustae* at 20 million spores/mL to certain non-target organisms, such as *Coturnix japonica*, *Apis mellifera*, *Bombyx mori*, *Daphnia magna*, and *Brachydanio rerio*, was examined in China. The results showed that there was no risk to honey bees per os at the maximum exposure dose tested, no deaths in the five animal species tested in the contact toxicity experiments, and *N. locustae* was relatively safe for nontarget beneficial organisms in the environment [33].

## 3. Pathogenicity

*N. locustae*, as an obligate parasite, reproduces in host target cells. Its infection involves the polar tube of the spore to inject its plasma into the target cells [5]. It causes high mortality in L&G. The locust's main target organ is the host's adipose tissue (fat body) [9]. *N. locustae* penetrates fat body cells and produces meronts, sporonts, sporoblasts, and spores. In the migratory locust, *Locusta migratoria migratorioides* (Reiche & Fairmaire, 1849), the younger the nymphs, the more susceptible they are, but even newly emerged adults are still susceptible [9]. Studies by Tounou et al. [34] on the effects of *N. locustae* on the desert locust, *Schistocerca gregaria* (Forskål, 1775) and the Senegalese grasshopper, *Oedaleus senegalensis* (Krauss, 1877), showed that *N. locustae* has high pathogenicity on the young nymphal instars of these two species. The median survival time for first, second, third,

fourth, and fifth nymphal instars was 6, 9, 10, 14, and 15 days, respectively, when locusts were inoculated with $1 \times 10^7$ spores on 10 g wheat bran in groups. Similar results were obtained with the Senegalese grasshopper using the same treatment method, with median survival times for instars 1, 3, and 5 being 5, 9, and 15 days, respectively. The median survival time therefore increased with the age of the locust and with decreasing inoculation doses. For instance, third-instar desert locust nymphs were inoculated in groups with $5.62 \times 10^6$ or $3.16 \times 10^4$ spores with a median survival time of 14 and 16 days, respectively. However, cumulative mortality increases with the increase of spore concentration and decreases with the increasing age of the nymphs. *N. locustae* caused high mortality in young desert locust nymphs. Mortality was 100% in first and second nymphal instars inoculated in groups with $1 \times 10^7$ spores of *N. locustae*, as well as in first nymphal instars with $1 \times 10^6$ spores. With third and fourth nymphal instars inoculated with $1 \times 10^7$ spores, cumulative mortality was above 90%. Similar results were obtained with the Senegalese grasshopper, which was inoculated with $1 \times 10^7$ spores: 100% mortality was observed in the first instar but 88.5% mortality in the third instar. In the fifth instar, mortality was only 66.3% in desert locust and 70% in the Senegalese grasshopper. These results suggest that the appropriate period for the application of *N. locustae* is mainly between the first and fourth instars, i.e., a possible application period of about 20–28 days, taking into account an average duration of 5–7 days per nymphal instar.

Inoculation of the 1st–5th nymphal instars of *Chondracris rosea* (De Geer, 1773) with *N. locustae* resulted in 78.2% to 100% mortality in field caging experiments by Liu and Chen [26]. The $LC_{50}$ was approximately $3.88 \times 10^5$ spores/mL for the 1st–3rd instars and $3.98 \times 10^6$ spores/mL for the 4th–5th instars. In a caging experiment on a large forest site, the same authors observed that *C. rosea* mortality was greater than 91.1% when they sprayed *N. locustae* at a rate of $5 \times 10^7$ spores/mL. At $1 \times 10^8$ spores/mL, mortality reached 100% 25 days after treatment [26]. Chen et al. [25] conducted a laboratory experiment with five concentrations of *N. locustae* spores ($1 \times 10^4$, $1 \times 10^5$, $1 \times 10^6$, $1 \times 10^7$, and $1 \times 10^8$ spores/mL) to treat early stages (1st–2nd instar) of the yellow-spined bamboo locust (*Ceracris kiangsu*). Mortalities were 16.6, 32.9, 29.2, 34.0, and 83.7%, respectively, increasing with spore concentrations. They also found that yellow-spined bamboo locust mortality was 85.1% after the application of a suspension of *N. locustae* spores at a concentration of $5 \times 10^7$ spores/mL in field trials in the forest. Zang et al. [35] reported that mortality of *Oxya chinensis* (Thunberg, 1815) was about 65.4–68.1% 30 days after the field treatment of third-instar nymphs sprayed with $1.5 \times 10^{10}$, $2.25 \times 10^{10}$ spores/ha, respectively. The infection rate in survivors was approximately 40%.

Concurrent use of *N. locustae* and *Metarhizium* spp. has shown additive effects on locusts and grasshoppers. When fifth-instar nymphs of *S. gregaria* were inoculated first with *N. locustae* at doses between $1 \times 10^4$ and $1 \times 10^6$ spores on wheat bran in groups, and then 10 days later with *M. anisopliae* at doses between $1 \times 10^2$ and $1 \times 10^4$ spores/nymph, the median survival times ranged from 3 to 9 days, and the shortest duration was only 3 days at doses of $1 \times 10^6$ spores of *Nosema* and $1 \times 10^4$ spores of *Metarhizium* [29]. When the locusts were inoculated in groups with *N. locustae* at doses of $1 \times 10^5$ and $1 \times 10^6$ spores, and *Metarhizium* at $1 \times 10^3$ and $1 \times 10^4$/nymph, the mortality was at 90% and 97% over 3 days, and both reached 100% 10 days after inoculation. Mortality was only 12.2% in the control, showing a synergistic effect between these two agents. Similar results were obtained with *N. locustae* directly mixed with *Metarhizium* spp., with the oriental migratory locust (*L. migratoria manilensis* Meyen, 1835). Mortality was higher with the mixture compared to treatments with each agent applied separately. The effects of a mixture of *M. acridum* and *N. locustae* on third nymphal instars under laboratory conditions showed that at a ratio of 1:1 (*M. acridum* at $3.90 \times 10^6$ spores/g locust body weight and *N. locustae* at $3.99 \times 10^6$ spores/g locust body weight), locust mortality was about 96.7% 24 days after inoculation, showing an additive effect of these two agents [36]. When 2nd–3rd-instar nymphs of *L. migratoria* were inoculated with $1 \times 10^6$ spores/nymph of *N. locustae*, and after 3, 6 and 9 days with *Metarhizium* at $1 \times 10^7$ conidia/mL, an additive effect was only

observed in nymphs inoculated with *Nosema* and then 9 days later with *Metarhizium* [37]. After examining the stage of *Nosema* development on Days 3, 6, and 9 after inoculation, they found that it was not until the *Nosema* spore maturation stage that locusts were most susceptible to fungal infection.

Lv et al. [38] identified 4 defensins from migratory locust palp transcriptomes, named LmigDEF1 (78 amino acids), LmigDEF3 (78 amino acids), LmigDEF4 (69 amino acids), and LmigDEF5 (67 amino acids) with theoretical isoelectric points (pI)/molecular weights (Mw, kDa) of 6.48/8.29, 6.56/8.37, 8.23/7.19, and 8.27/6.93, respectively. The expression patterns of LmigDEF1, LmigDEF3, and LmigDEF5 in the fat body and salivary glands were examined by qRT-PCR after the locusts were inoculated with *N. locustae*. Results indicated that all three defensins varied over time in the fat body and salivary glands after *Nosema* infection, the transcript level of the LmDEFs being at their lowest in the fat body over the 10 days. This indicates that *Nosema* infection reduced the locust's immune response via the defensins and may explain why the coordinated use of *Nosema* and *Metarhizium* results in higher locust mortality [29,36]. In particular, for about four days when *Nosema* were sporulating after inoculation, the locusts were more easily infected by the fungus due to the lower level of the three types of defensins in the fat body [37]. In addition, Chen et al. [39] demonstrated the key role of *N. locustae* sporulation in locust mortality. They identified a spore wall protein, AlocSWP2 from *N. locustae*, containing four cysteines. AlocSWP2 has been detected in the wall of mature spores, sporoblasts, and sporonts during sporulation in the host body by immunocytochemistry localization experiments. AlocSWP2 was detected in the fat body of infected locust only on Day 9 after inoculation using RT-PCR. The survival percentage of infected locusts that received a dsRNA injection of AlocSWP2 on Days 15, 16, and 17 after inoculation of *Nosema* spores was significantly higher than that of infected locusts without dsRNA treatment. Similarly, the number of spores in locusts infected with *Nosema* and treated with RNAi of AlocSWP2 was significantly lower than that in infected locusts without RNAi of this gene. This indicates that this *N. locustae* spore wall protein is involved in sporulation, contributing to host mortality.

*L. migratoria migratorioides* infected with *N. locustae* demonstrated reduced sustainable flight capacity [9]. Zhang et al. [40] confirmed this with the oriental migratory locust (*L. migratoria manilensis*). Flight capacity of infected and healthy adult locusts, 5 to 15 days after emergence, was determined using a flying mill for 18 h. On average, the flight distance of healthy versus infected locusts was 14,279 vs. 864 m; flight speed, 1.23 vs. 0.51 m/s, flight time, 3 vs. 0.33 h; maximum flight distance, 72,538 vs. 1544 m; and maximum sustained flight time, 6.7 h vs. 0.1 h. The decrease in flight capacity may be due to the fact that *N. locustae* destroyed the fat bodies reducing the supplement of glyceride and fat as energy resources. In *L. migratoria manilensis* infested with *N. locustae*, the glyceride content decreased rapidly, while lipase activity increased in both hemolymph and total fat [41].

Locusts and grasshoppers infected with *N. locustae* have reduced fertility [13,42]. Reduced vitellogenin was observed in fourth-instar nymphs of *L. migratoria manilensis* inoculated with *N. locustae*: vitellogenin levels in the fat body, hemolymph, and ovaries were very low compared to the control [43]. The maximum vitellogenin level in infected vs. healthy locusts, respectively, were 4.663 vs. 18.655 mg/mL in the fat body, 2.627 vs. 7.603 mg/mL in the hemolymph, and 4.927 vs. 73.367 mg/mL in the ovaries. This explains why the reproductive capacity of infected locusts is low.

The disease caused by *N. locustae* is transmitted vertically in eggs and egg pods [5]. Infected females have been reported to lay eggs containing spores of *N. locustae* [42]. Raina et al. [44] reported vegetative stages of *N. locustae* in the yolk of the oocyte and spores in the eggs of *L. migratoria*. When fourth-instar nymphs were inoculated with a dose of $1.5 \times 10^6$ spores, the infected parents laid eggs and the prevalence of infection was 100% in the next generation. The disease was vertically transmitted up to 14 generations, and mortality due to vertical transmission at times reached more than 90%. Parents of *S. gregaria* and *O. senegalensis* infected with *N. locustae* produced progeny with a 50% infection rate, indicating high vertical transmission in these species [34].

## 4. Genomics and Molecular Biology

Understanding the molecular biology of *N. locustae* is fundamental in determining its biology and molecular interactions with the host and improving applications in its use for pest control. A complete genomic sequence of *N. locustae* has recently been revealed by Chen et al. [45], which is a major achievement. Sequencing of its genome yielded 3,170,203 nucleotides, encoding for 1857 predicted genes, of which 1755 are single-exon genes, and 102 are multiple exon genes. A total of 17 scaffolds, ranging from 88.763 to 388.82 kb, were identified and assigned to 17 chromosomes. Genomic and protein sequences from *N. locustae* and several other single-celled organisms were used to study evolutionary relationships in genetic synteny and collinearity using the MCScanX method. Results showed that within the microsporidia, most genes exhibit good collinearity [45].

In addition to genomic sequencing, Chen et al. [45] performed an analysis of locust midgut transcripts versus locust fat body transcripts from *N. locustae* infected locusts. They found that the abundant expression of locust antimicrobial peptides and other defense genes, such as peroxiredoxin and amine oxidase in the midgut, may explain the lower number of microsporidian spores in the midgut of the host. In the fat body, however, the large number of *N. locustae* spores present may be related to the fact that several locust phenol oxidases and peroxisome proliferator-activated receptors have been inhibited to allow it to escape the host immune response. This revealed the interactions of *N. locustae* and the host at the transcript molecular level, in particular why *N. locustae* can reproduce massively in the locust fat body and not in the gut.

Identification of the genes and proteins of the polar tube is certainly useful in understanding the interactions between *N. locustae* and its hosts at the molecular level. Two polar tube protein genes, PTP1 and PTP2, have been identified, and it has been suggested that PTP1 plays a key role in interactions with the host cell surface [46]. Two other orthologous polar tube proteins, named AlPTP2b and AlPTP2c, exhibit elastomeric characteristics. The AlPTP2b and AlPTP2c genes encode for proteins of 568 and 599 amino acids with deduced molecular weights of 55,399 and 56,664 Da, respectively. These proteins are highly conserved (84.2% identity), larger than the previously reported AlPTP2 (287 amino acids) [47].

Fast et al. [48] identified a gene for the TATA box binding protein (TBP) of *N. locustae*. The predicted amino acid sequence of the TBP gene consists of 259 amino acids. In the phylogenetic analysis of TBP, the authors emphasized that TBP from *N. locustae* is close to that of fungi and supported previous studies on the evolution of microsporidia with tubulins, HSP70, and the greater subunit of RNA polymerase II (RPB1) proteins [49–53]. However, in a study on the large subunit (LSU) rRNA, Peyretaillade et al. [54] proposed that the origin of microsporidia was not specifically linked to a particular group of eukaryotes. In general, there are only a few studies on proteins and genes of *N. locustae* [55], and even rarer studies on their functions, especially pathogen–host interactions at the molecular level.

## 5. Mass Production and Products

The production of *N. locustae* spores is done in vivo. The rearing of infected hosts is the main process for mass production (Figure 2). Grasshoppers *Melanoplus bivittatus* (Say, 1825) have been used as hosts in the United States. Henry et al. [10] pointed out that several factors influence production yield. Cage size and the number of individuals in each cage, light in the cage, and time of harvest, as well as factors such as grasshopper species and sex, are important. The general process is as follows: grasshopper nymphs are reared to the 4th–5th instars, inoculated with *N. locustae* spore suspension, and then reared until they die. The cadavers are collected and crushed, screened, and centrifuged to obtain a high concentration of spores, which are stored between $-10$ and $-20\ ^\circ$C. In the United States, after several improvements in mass production techniques, a yield of $1 \times 10^9$ to $3 \times 10^9$ spores per grasshopper has been achieved. However, the use of *Melanoplus differentialis* (Thomas, 1865) as a host resulted in up to $7.1 \times 10^9$ spores per

individual [56]. The time of harvest is an important factor in influencing the spore yield; the further away from the inoculation date, the more spores are obtained [17].

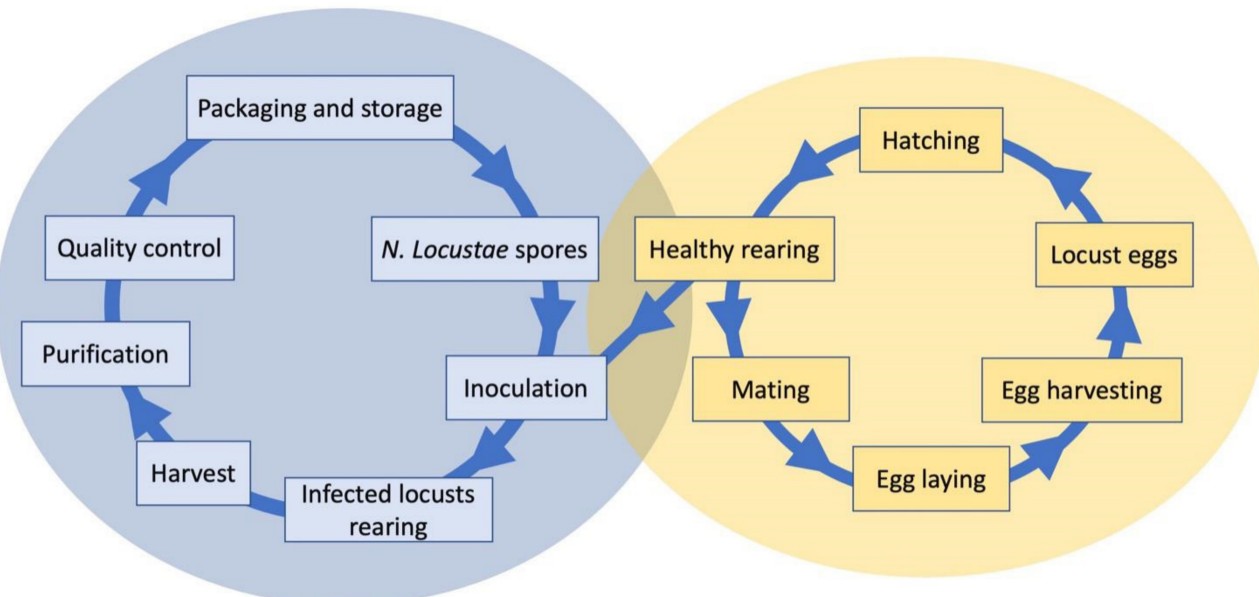

**Figure 2.** Mass producing process of *N. locustae* spores.

In China, *L. migratoria manilensis* has been selected as a host to produce spores of *N. locustae*. In this species, there is no diapause, and individuals can be used year-round. Several factors influence the efficiency of mass production, including the inoculation concentration, the stage of development for inoculation, and the harvesting time [57]. A higher concentration of spores in the inoculation suspension resulted in rapid locust death, and a lower concentration increased the time to harvest and reduced the number of spores. The best inoculation concentration was found to be $1 \times 10^6$ spores/mL, the stage for inoculation the fourth instar, and the harvest time about 30–40 days after inoculation. The average spores/individual yield was approximately $6 \times 10^9$, the highest being about $9.9 \times 10^9$. In China, several insectariums have been established for the mass production of *N. locustae* since the 1990s. Currently, the average spore yield per individual can reach $22 \times 10^9$ to $34 \times 10^9$. There are about 5–6 harvests per year, and the total yield can reach $2 \times 10^{15}$ spores for each insectarium [17].

The products are usually formulated either as bran bait, aqueous suspension, or water-based suspension. They are prepared from high concentrations of spores stored at low temperature, diluted directly with water or mixed with wheat bran in an appropriate ratio. In China, a new water-based suspension has been developed. The spores are mixed with xanthan gum, sorbic acid, and other environmentally friendly additives. The suspension is more stable and homogeneous and can be stored for about one year at room temperature [58]. It is better adapted to the needs of users and should be widely applied. To date, several commercial products have been registered such as Nolo Bait™, Semaspore™, and Grasshopper Attack™ in the United States since the 1980s [12]. In China, *Nosema locustae* products for locust control have been developed since the 1990s and are currently distributed by Beijing JiaJing Biotechnology Ltd. (Beijing, China). They are based on a highly pathogenic strain (AL2008L-04) that significantly improves product efficacy, a high-yield production technology, and an aqueous suspension formulation that can be stored at room temperature [59].

## 6. Application and Epizootics

The three main components whose interactions can lead to a reduction in locust and grasshopper populations and epizootics are *N. locustae,* its hosts, and its environments. *N. locustae* has been used to control *L. migratoria manilensis* since the 1990s. A field trial was conducted on Hainan Island, China, in areas close to rice fields [60]. The locust population consisted of first to third instars with pretreatment densities of 0.9, 1.5, and 2.2 individuals/m$^2$ in the three treatment plots and 0.6 in the control plot. The plots were treated at $50 \times 10^9$, $99 \times 10^9$, and $150 \times 10^9$ spores/ha. At 25 days after treatment, the locust population density was reduced by 70.2%, 74.0%, and 78.9%, respectively, in the treated plots but increased by 58.1% in the control plot. The prevalence of *N. locustae* infection in survivors was 29.4%, 31.1%, and 31.8% at 30 days and 40.6%, 40%, and 35.7% at 40 days after treatment in the treated plots. The prevalence of infection persisted the following summer (one year) in the three treated plots, respectively, at 23.5%, 13%, and 9.1% levels.

In the United States, in Montana, studies indicated that the prevalence of *N. locustae* during the treatment season was variable, but mainly around 30%, with the highest being 50% when the spore application rate was $1.4 \times 10^{10}$ in 1.12 kg of wheat bran/ha after three applications [10,61]. In Canada, in a grasshopper population consisting mainly of third instars of *Melanoplus sanguinipes*, *M. packardii* (Scudder, 1878), and *Camnula pellucida* (Scudder, 1862) treated with $2.5 \times 10^9$ or $5.0 \times 10^9$ spores in 1.68 kg wheat bran per ha, an infection rate of approximately 50% was observed 4–5 weeks after treatment [62]. Toward the end of the season, about 12 weeks after treatment, this rate was about 95–100%. Zhang et al. [63] reported the prevalence of *N. locustae* in the years following the application of spore-based baits to mixed grasshopper populations—consisting of *Oedaleus asiaticus* Bey-Bienko, 1941, *Myrmeleotettix palpalis* (Zubovski, 1900), *Angaracris rhodopa* Fischer von Waldheim, 1836, *A. barabensis* (Pallas, 1773) and *Dasyhippus barbipes* (Fischer von Waldheim, 1846)—in Inner Mongolia rangelands, a region where temperatures are below −10 °C in winter and 20 to 30°C in summer and rainfall is about 200 mm per year. In two areas treated in 1991 and 1992 with $7.5 \times 10^9$ spores/1.5 kg wheat bran bait/ha, locust infection rates of *N. locustae* were 14.5% and 19.8% in the second year after treatment. In a third area treated in 1988 with *N. locustae* baits, the observed infection rates were 6.8% and 23% two and six years after treatment. Mortality rates of 60–80% in mixed grasshopper populations were achieved following applications of *N. locustae,* in Inner Mongolia rangeland, at a rate of $3 \times 10^{10}$ spores/ha for two consecutive years [64]. *N. locustae* was even used in the highlands (3270–3350 m ASL) in Qinghai province of China, and the disease could still be transmitted and persisted for a long time among grasshoppers [65], up to 10 years with an infection rate exceeding 50% in some years [27]. In three locations in Canada in two consecutive years (1988–1989), Johnson and Dolinski [66] found that *N. locustae* persisted in grasshopper populations. They suggested that the activity of *N. locustae* was not inhibited despite severe weather conditions, hot in summer and very cold in winter, with air temperature down to −39 °C.

*N. locustae* was reportedly applied in a forest to control the yellow-spined bamboo locust (*Ceracris* kiangsu) [25]. In a 1.5 ha area of bamboo sprayed with 50 kg of *N. locustae* spore suspension at a concentration of $5 \times 10^7$ spores/mL, mortality of *C. kiangsu* was 85% 15 days after treatment. *N. locustae* was also used in 2018 to control the yellow-spined bamboo locust in bamboo forest ecosystems in Phongsaly province, Lao [67]. Nymphs were in the third and fourth instars at a density of over 100 individuals/m$^2$. Treatment was carried out on May 13 at an application rate of $2 \times 10^7$ spores/mL. After 17 days, many dead locusts were observed in the treated area, where the average density was less than 10 individuals/m$^2$, a reduction of more than 90% compared to the untreated control area. *N. locustae* spores were observed in all dead locusts that were collected from the treated plots and examined individually under a microscope in the laboratory. The average grade of infection was three out of five, corresponding to severe disease. In contrast, the infection rate of locusts collected from the control area was zero. In addition, 46% of surviving

locusts in treated plots were infected with *N. locustae*, and two and a half months after treatment, the density was very low, about 0–2 individuals/m$^2$. It was concluded that *N. locustae* could be an effective agent to control high densities of yellow-spined bamboo locusts in bamboo forests.

In Africa, Tounou et al. [68] conducted field trials to assess the effects of *N. locustae* and *Metarhizium anisopliae* against various grasshopper species: *Pyrgomorpha cognata* Krauss, 1877, *Acrotylus blondeli* Saussure, 1884 (both predominant species), and *Oedaleus senegalensis*. Both agents were mixed with wheat bran as bait with *M. anisopliae* alone, *N. locustae* + *M. anisopliae*, *N. locustae* spores alone, or *N. locustae* + sugar. The same treatments were carried out with *O. senegalensis* in the early stages of development and resulted in mortalities of 64–85%. The population density during the three weeks of monitoring decreased by 44.7 ± 6.9% in the *N. locustae* plot, 52.8 ± 8.4% in the *N. locustae* + sugar plot, 73.7 ± 5.5% in the *M. anisopliae* plot, and 89.1 ± 1.8% in the *N. locustae* + *M. anisopliae* plot. The prevalence of *N. locustae* in adult grasshoppers surviving at 28 days after application was 48.1 ± 2.3, 28.9 ± 4.8, and 27.4 ± 3.7% in the three treatments with *N. locustae*. The results suggest that these two biological control agents have the potential to control the early stages of grasshoppers in Africa. These conclusions differ from those of Lima et al. [69], who found no significant difference between treated and untreated plots and concluded that *N. locustae* could not be used to control *O. senegalensis* in the Cape Verde archipelago. However, they did not specify the developmental stage of the treated grasshoppers, and it is possible that efficacy was less on the fifth nymphal instar or on adults.

Long-term epizootics of *N. locustae* have been investigated in Argentina by Lange et al., who have shown that this pathogen can persist for many years after its introduction [30,70,71]. These authors, in their field survey, found that the prevalence of *N. locustae* ranged from 1.8 to 41% in 9 of 13 sites between 1995 and 2003. In 1990, in another field survey in an area treated 11 years earlier, Lange and Azzaro [71] observed infection rates of 2.9%, 3.5%, and 3.6% by *N. locustae* spores in three grasshopper species, *Dichroplus elongatus* Giglio-Tos, 1894, *D. maculipennis* (Blanchard, 1851), and *Scotussa lemniscata* (Stål, 1861). Lange et al. [30] examined grasshoppers collected from sites treated with *N. locustae* in Argentina in the late 1970s, early 1980s, and mid-1990s and found that the highest prevalence of *N. locustae* in grasshopper populations was about 50%.

## 7. Conclusions and Prospects

*N. locustae* is becoming increasingly biologically understood, as well as for its potential in L&G control. Recent studies have shown that it can be applied in different ways to achieve control objectives and plays a more important role than ever in locust and grasshopper management programs. It has a fairly broad host spectrum, and at least 144 orthopteran species are susceptible. However, it is very safe for non-orthopteran insects and other non-target animals [17,33]. It is transmitted orally and is therefore probably less sensitive to environmental factors than fungal-type agents. This is certainly the reason why good results have been recorded at a 3000–4000 m altitude on the Tibetan plateau in China and in high-temperature regions such as Hainan province in China or Lao and Vietnam. *N. locustae* has been demonstrated to be efficient in the control of L&G in rangelands, crop fields, and forest ecosystems. Its disease can persist for many years after application in L&G populations and can be part of their long-term management.

When L&G are at low density, a low dose of *N. locustae* will slowly kill them as a long-term control method. However, recent field trials have indicated that *N. locustae* can also be used at high locust densities, as has been demonstrated to control second and third instars of the yellow-spined bamboo locust. *N. locustae* also exhibits a synergistic effect when used in admixture with the fungus *Metarhizium* spp. and can potentially weaken the host immune response by reducing its defensins. The simultaneous use of *N. locustae* and *Metarhizium* spp. could be encouraged to control high-density locust outbreaks.

L&G mortality caused by *N. locustae* is dose-dependent; the higher the dose used, the higher the mortality. However, *N. locustae* spores are produced restrictively in vivo. The

mass rearing of locusts to produce spores is expensive; therefore, increasing the spore yield per locust or grasshopper is a bottleneck for mass production and large-scale application. Understanding the molecular mechanisms of *N. locustae* and its interactions with its host may be valuable to explore ways to mass-produce *N. locustae* spores in order to improve the traditional in vivo mass production method or to find an alternative, as well as to increase the effects on L&G. Genome sequencing and transcriptomic analysis of *N. locustae* genes and proteins have provided a valuable basis for future studies to improve the genetics of *N. locustae* and screen for high-virulence strains [72]. The development of new formulations and other technologies to maintain high spore vitality at room temperature will also be useful in promoting the wider application of *N. locustae*.

**Author Contributions:** L.Z. conceived the article and drafted the manuscript with substantial input from M.L. All authors have read and agreed to the published version of the manuscript.

**Funding:** This work is supported by a grant from Shandong Academy of Agricultural Sciences, and a grant from the Ministry of Agriculture of China (15206030).

**Acknowledgments:** Authors would like to thank anonymous reviewers' comments and suggestions.

**Conflicts of Interest:** The authors declare no conflict of interest.

**Disclaimer:** Mention of trade names or commercial products in this publication is solely for the purpose of providing specific information and does not imply recommendation or endorsement by the authors.

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
