# Peer review of "Nosema locustae (Protozoa, Microsporidia), a Biological Agent for Locust and Grasshopper Control"

_agronomy, doi:10.3390/agronomy11040711_

Round 1

Reviewer 1 Report

This a timely review that brings together the past and more importantly the recent developments in the study of this pathogen as a potential agent for management of grasshoppers and locusts. However I found the text difficult to follow at times and have made many suggestions for changes using the track changes function of Word. As well I have entered comments directly on the manuscript.

The reading was at times confusing especially when trying to follow which statements are attributed to which citations. Also, there is no need to provide too much detail. For instance, would it not suffice to simply state that infection is dose dependant without providing all of the details of which concentration elicited which percent infection? Also, providing a spore concentration as number of spores per ml is meaningless unless details are also provided on how this dose was administered as number per nymph or surface area.

I suggest that when a new grasshopper or locust is first mentioned, it be presented as species name followed by common name. Thereafter only species name should be used. Interchanging species and common names throughout is confusing.

Would not the host range be better presented as a table. I current section on this is very difficult to follow and get a grasp of the situation.

A section needs to be devoted on pathogenesis. How is the infection initiated? How and when are spores produced, etc. This information seems to appear here and there throughout the review; however, it needs to be covered in detail in a separate paragraph or 2.

More suggestions are entered on the manuscript.

Author Response

This a timely review that brings together the past and more importantly the recent developments in the study of this pathogen as a potential agent for management of grasshoppers and locusts. However I found the text difficult to follow at times and have made many suggestions for changes using the track changes function of Word. As well I have entered comments directly on the manuscript.

Answer: Thank you very much to have revised our manuscript and a lot very valuable suggestions and comments! We have changed the places as your suggestions.

The reading was at times confusing especially when trying to follow which statements are attributed to which citations.

Answer: The citations have been changed as what you pointed out.

Also, there is no need to provide too much detail. For instance, would it not suffice to simply state that infection is dose dependant without providing all of the details of which concentration elicited which percent infection?

Answer: Some sentences have been cancelled.

Also, providing a spore concentration as number of spores per ml is meaningless unless details are also provided on how this dose was administered as number per nymph or surface area.

Answer: As Nosema spores are large and have large sedimentation coefficient, when the spore suspension is used, almost all of the spores are deposited on the surface of plants leave in the field trials. In the laboratory experiments, some spores were treated in dosage for each locust, some in concentration.

I suggest that when a new grasshopper or locust is first mentioned, it be presented as species name followed by common name. Thereafter only species name should be used. Interchanging species and common names throughout is confusing.

Answer: That is a good way. We have changed according to your suggestion.

Would not the host range be better presented as a table. I current section on this is very difficult to follow and get a grasp of the situation.

Answer: We have used a table to show the host range (Table 1).

A section needs to be devoted on pathogenesis. How is the infection initiated? How and when are spores produced, etc. This information seems to appear here and there throughout the review; however, it needs to be covered in detail in a separate paragraph or 2.

Answer: Good idea. We have included a paragraph on pathogenicity.

More suggestions are entered on the manuscript.

Answer: We have changed according to your very valuable suggestions and comments.

As for the question: “introducing commercial names”.

Answer: However, we authors think that it is not suitable to provide commercial names in article. There are three commercial products in China, as active ingredients as 10X109 spores/ml, 2X107sopres/ml, 4X107 spores/ml, in US such as NoLo BaitTM.

Reviewer 2 Report

This review aims to summarize the work and understanding of Nosema locustae as a biocontrol agent against various locust and grasshopper species. While I appreciate that a summary and synthesis of this work may be valuable, the authors have not executed this mission to my satisfaction. The writing is verbose, under-synthesized, and poorly edited. The work is riddled with typos (too many spaces, forgetting spaces, inconsistent formatting, etc.). Overall, I feel that this review suffers from underdevelopment. The authors need to take a critical look at their writing, synthesis, and presentation of the work they aim to summarize. I feel there is an overemphasis of work done in China and on Chinese Acridids. Important work from South America is mentioned briefly in comparison and the work of prominent microsporidian experts, like Leellen Solter, is largely absent. The writing is underdeveloped and often feels like a laundry list of papers and overly detailed reexplanations of published works rather than critical analysis and synthesis. The authors should think about assembling some take-away figure, model, or table that summarizes the work in this field in a new way. Finally, the authors conclude this work with an emphasis on an improved formulation for shelf-life and storage longevity, but hardly mention this elsewhere in the paper. This work feels rushed and, in its present form, is not contributing much to the field. 

Author Response

Reviewer 2:
R-This review aims to summarize the work and understanding of Nosema locustae as a biocontrol agent against various locust and grasshopper species. While I appreciate that a summary and synthesis of this work may be valuable, the authors have not executed this mission to my satisfaction. The writing is verbose, under-synthesized, and poorly edited. The work is riddled with typos (too many spaces, forgetting spaces, inconsistent formatting, etc.).

Answer: Thank you for comments. We have revised the manuscripts.

R - Overall, I feel that this review suffers from underdevelopment. The authors need to take a critical look at their writing, synthesis, and presentation of the work they aim to summarize. I feel there is an overemphasis of work done in China and on Chinese Acridids. Important work from South America is mentioned briefly in comparison and the work of prominent microsporidian experts, like Leellen Solter, is largely absent.

Answer:  Based on your comments we have reorganized the paragraphs, sentences, and added some summary sentences. Yes, in this manuscript, there seem to be more articles on the use of N. locustae in Asia. Recently, China has indeed developed more than other parts of the world in the use of this pathogen. However, this manuscript does not focus too much on work in China and Chinese work. In fact, the work in Africa and Argentina is also highlighted. Thank you for this good suggestion. We have added the work of Solter and some other authors.

R - The writing is underdeveloped and often feels like a laundry list of papers and overly detailed reexplanations of published works rather than critical analysis and synthesis. The authors should think about assembling some take-away figure, model, or table that summarizes the work in this field in a new way. Finally, the authors conclude this work with an emphasis on an improved formulation for shelf-life and storage longevity, but hardly mention this elsewhere in the paper. This work feels rushed and, in its present form, is not contributing much to the field.

Answer: We have revised the manuscript, and added a photo, a figure and a table based on your suggestions. Our manuscript reviews recent works on studies and applications of N. locustae as a biological control agent for locusts and grasshoppers, which is useful in promoting further using and understanding of N. locustae.

Round 2

Reviewer 1 Report

The manuscript is certainly easier to read and the authors have adequately addressed most of my previous concerns. However more cleaning up is in order according to the following line numbers:

19 insert “and” before “is well adapted”

40 font change

48 insert “and” before “epizootiology”

54 font change

63 is it necessary to give photo credit if the credit goes to one of the authors?

91 font change

109 1st not 1th here and throughout

120 1st and 3rd .. .4th -5th instars here and throughout.

135 & 139  groups (plural)

267 delete “to  produce N. locustae spores.” This is understood, no need to state.

273 must provide a unit of volume if this is a concentration. Per ml I presume.

287 I still this it is important to indicate what these products are and produced by whom.

300 insert “and” before 31.8 and before 35.7.

320 delete “respectively” not necessary here.

331 font change

358 here and throughout delete unnecessary bolding e.g. 367, 372, 373….

358 “respectively” is not used correctly here. Delete.

363 replace “these authors” with “they”

381 “Orthoptera” is the proper name. orthopterans is not so no need to capitalize the “O”.

381 delete “insects”  All orthopterans are insects.
